# Anti-Inflammatory Effects of GPR55 Agonists and Antagonists in LPS-Treated BV2 Microglial Cells

**DOI:** 10.3390/ph17060674

**Published:** 2024-05-24

**Authors:** Lu Sun, Matthias Apweiler, Claus Normann, Christoph W. Grathwol, Thomas Hurrle, Simone Gräßle, Nicole Jung, Stefan Bräse, Bernd L. Fiebich

**Affiliations:** 1Neuroimmunology and Neurochemistry Research Group, Department of Psychiatry and Psychotherapy, Medical Center-University of Freiburg, Faculty of Medicine, University of Freiburg, D-79104 Freiburg, Germany; lu.sun@uniklinik-freiburg.de (L.S.); matthias.apweiler@uniklinik-freiburg.de (M.A.); 2Department of Psychiatry and Psychotherapy, Medical Center-University of Freiburg, Faculty of Medicine, University of Freiburg, D-79104 Freiburg, Germany; claus.normann@uniklinik-freiburg.de; 3Institute of Biological and Chemical Systems-Functional Molecular Systems (IBCS-FMS), Karlsruhe Institute of Technology (KIT), Kaiserstrasse 12, D-76131 Karlsruhe, Germany; christoph.grathwol@kit.edu (C.W.G.); hurrle.thomas@gmail.com (T.H.); simone.graessle@kit.edu (S.G.); nicole.jung@kit.edu (N.J.); braese@kit.edu (S.B.); 4Institute of Organic Chemistry, Karlsruhe Institute of Technology (KIT), Kaiserstrasse 12, D-76131 Karlsruhe, Germany

**Keywords:** neuroinflammation, coumarin derivative, GPR55, O-1602, ML-193, cytokines, chemokines

## Abstract

Chronic inflammation is driven by proinflammatory cytokines such as interleukin 6 (IL-6), tumor necrosis factor-α (TNF-α), and chemokines, such as c-c motif chemokine ligand 2 (CCL2), CCL3, C-X-C motif chemokine ligand 2 (CXCL2), and CXCL10. Inflammatory processes of the central nervous system (CNS) play an important role in the pathogenesis of various neurological and psychiatric disorders like Alzheimer’s disease, Parkinson’s disease, and depression. Therefore, identifying novel anti-inflammatory drugs may be beneficial for treating disorders with a neuroinflammatory background. The G-protein-coupled receptor 55 (GPR55) gained interest due to its role in inflammatory processes and possible involvement in different disorders. This study aims to identify the anti-inflammatory effects of the coumarin-based compound KIT C, acting as an antagonist with inverse agonistic activity at GPR55, in lipopolysaccharide (LPS)-stimulated BV2 microglial cells in comparison to the commercial GPR55 agonist O-1602 and antagonist ML-193. All compounds significantly suppressed IL-6, TNF-α, CCL2, CCL3, CXCL2, and CXCL10 expression and release in LPS-treated BV2 microglial cells. The anti-inflammatory effects of the compounds are partially explained by modulation of the phosphorylation of p38 mitogen-activated protein kinase (MAPK), p42/44 MAPK (ERK 1/2), protein kinase C (PKC) pathways, and the transcription factor nuclear factor (NF)-κB, respectively. Due to its potent anti-inflammatory properties, KIT C is a promising compound for further research and potential use in inflammatory-related disorders.

## 1. Introduction

Microglia are the resident macrophages and main immune cells of the central nervous system (CNS) [1], serving as the frontline defense against exogenous toxic substances and proinflammatory events [2]. However, they can also release proinflammatory mediators and induce inflammatory processes in response to stimuli [3]. Chronic low-grade neuroinflammation by overactivated microglia is a key factor in the development of neurodegeneration, resulting in neuropsychiatric and cognitive symptoms. Activated microglia accelerate neuroinflammatory and neurotoxic responses by releasing various proinflammatory cytokines, including interleukin-6 (IL-6), tumor necrosis factor-α (TNF-α), and chemokines, such as c-c motif chemokine ligand 2 (CCL2), CCL3, C-X-C motif chemokine ligand 2 (CXCL2), and CXCL10 [4]. These neuroinflammatory responses are correlated with neurodegenerative diseases and psychiatric disorders such as Alzheimer’s disease (AD), Parkinson’s disease (PD), and depression [5].

The inflammatory hypothesis of AD proposes that cytokines spark an acute-phase response as a relevant part of AD pathogenesis [6]. Anti-inflammatory drugs might reduce the incidence of AD and could be beneficial in the treatment of AD [7]. Non-steroidal anti-inflammatory drug (NSAID) exposure is correlated with a reduced risk for AD in different studies; however, there are several limitations, and the strength of the association is quite weak [8]. Increased levels of CXCL10, in addition to some other chemokines, have been reported in blood samples of AD patients compared to healthy controls in a meta-analysis covering 23 studies [9]. In PD, NSAID exposure was correlated with a decreased risk for PD only in a couple of studies, but NSAIDs might boost the dopaminergic treatment in PD as co-therapeutics [10]. In addition to neurodegenerative diseases, inflammatory processes are associated with psychiatric disorders, such as depression. Increased levels of proinflammatory mediators and markers, such as CCL2, CRP, and IL-6, were found in patients with severe depression [11]. Effective treatment led to the normalization of those markers [11], supporting the contribution of inflammation to the symptoms of depression [12].

Furthermore, anti-inflammatory agents enhanced the effects of antidepressant treatment in a meta-analysis [13]. In major depression, lower concentrations of CCL2 in blood serum were observed when compared to healthy controls. Interestingly, the use of antidepressants did not alter chemokine levels in this study [14]. In depressive suicidal patients, a decreased expression of CCL2 and CXCL2 in the prefrontal cortex was found, while expression of CXCL10 was not altered [15]. However, higher levels of CCL2 and CXCL10 were found in murine brains after induction of depressive-like behavior by administration of LPS or interferon α (IFNα) [16].

To sum up, the involvement of chemokines in neurological and psychiatric disorders is still the subject of ongoing research, and available studies show contrary results. Overall, effective pharmacotherapies in neurodegenerative and psychiatric disorders are limited, and their mechanisms of action are still rudimentarily understood. Therefore, ongoing research still focuses on those disorders’ pathophysiology, involvement of anti-inflammatory processes, and development of novel anti-inflammatory compounds with less harmful side effects.

The orphan G-protein-coupled receptor 55 (GPR55) is highly expressed in the CNS [17] and in peripheral tissues [18]. A growing body of research demonstrates the expression of GPR55 in microglia [19] and its involvement in inflammation [20]. However, the role of GPR55 in the pathophysiology of inflammation is still the subject of current research due to contrary findings. The binding of an extracellular ligand to the GPR55 results in signal transduction via the G proteins Gα_12/13_ [21] and Gα_q_ [22], leading to the phosphorylation and activation of phospholipase C (PLC) and protein kinase C (PKC). The activation of GPR55 results in an increase in intracellular Ca^2+^ concentrations as well as phosphorylation of extracellular signal-regulated kinases 1/2 (ERK 1/2) and p38 mitogen-activated protein kinase (MAPK) [23]. The kinases lead to the activation of transcription factors, such as cAMP-response element-binding protein (CREB), nuclear factor κ-light chain enhancer of activated T cells (NF-κB) [24], and Akt serine/threonine protein kinase [25].

Several commercially available GPR55 agonists, such as O-1602, and GPR55-antagonists, like ML-193, are widely used in GPR55 research [23]. O-1602 enhanced the release of the proinflammatory cytokines IL-12 and TNF-α and decreased the endocytic activity of lipopolysaccharide (LPS)-stimulated monocytes and natural killer (NK) cells [20]. In contrast, it has been demonstrated that the GPR55 agonist O-1602 decreased IL-6 and TNF-α in an acute pancreatitis model [26]. ML-193, as a GPR55 antagonist, decreased LPS-induced prostaglandin E_2_ (PGE_2_) release and elicited anti-neuroinflammatory effects in primary microglia [27]. Modulation of depressive-like behavior in animal models, as well as modulation of inflammatory responses using commercially available GPR55 agonists and antagonists, has been described [28,29,30,31]. However, studies using GPR55 agonists and antagonists have reported both pro- and anti-inflammatory effects for the same ligands. As discussed before, biased agonism might be responsible for the observed differentiated effects in addition to the different cells or species being used [32]. The role of GPR55 in the regulation of inflammation, therefore, remains the focus of current research. 

In addition to the commercially available GPR55 agonists and antagonists, several natural compounds with a coumarin scaffold exert anti-inflammatory activities [33]. One possible mechanism of those anti-inflammatory effects is an inverse agonistic activity at GPR55, which is observed for coumarin derivatives with chemical residues at C3 and C8 showing high GPR55 affinity, while chemical residues at C7 boost cannabinoid receptor (CB) affinity while reducing GPR55 affinity [34]. In previous studies, we have shown that the coumarin derivative KIT C decreased PGE_2_ release and revealed anti-oxidative effects in IL-1β-stimulated SK-N-SH cells as well as in LPS-stimulated primary mouse microglia, which were abolished after GPR55 knock-out in SK-N-SH neuroblastoma cells [32,35]. Furthermore, we were able to identify anti-neuroinflammatory effects of various coumarin-based compounds KIT 10 [27], KIT 3, KIT 17, and KIT 21 [19] in primary microglial cell cultures, which also showed antagonistic binding affinities to GPR55. Those KITs act as GPR55 antagonists with an inverse agonistic activity. 

Based on our studies with those coumarin derivatives and especially KIT C in primary rat and mouse microglia [19,27,32,36], we further investigated the effects of KIT C, the commercially available GPR55 agonist O-1602, and the GPR55 antagonist ML-193 (Figure 1) on LPS-stimulated expression and the release of different cytokines and chemokines in BV2 microglial cells in this study. This cell line was used to gain additional data on the modulation of GPR55-associated signal pathways by treatment with three GPR55 ligands. This study aims to expand the understanding of the anti-inflammatory mechanisms of GPR55. KIT C was chosen due to its promising anti-inflammatory effects in LPS-stimulated primary mouse microglia (IL-6 and PGE_2_) when compared to KIT H.

## 2. Results

### 2.1. Effects of KIT C, ML-193, and O-1602 on Cell Viability

We have previously shown that the coumarin derivative KIT C inhibits the release of PGE_2_ in primary mouse microglia, demonstrating its anti-inflammatory effects in primary cell cultures. To avoid the additional use of animals for further neuroinflammation studies investigating this compound, we switched to cultured BV2 microglial cells in this study, a well-established microglial cell line sharing various characteristics with primary microglia, especially in the synthesis of inflammatory molecules such as chemokines and cytokines after stimulation with LPS [37]. Here, we analyzed the effects of KIT C, ML-193, and O-1602 on LPS-mediated expression and synthesis of the cytokines TNF-α and IL-6 and on the chemokines CCL2, CXCL2, CCL3, and CXCL10.

We first analyzed cell viability of BV2 microglial cells after treatment with KIT C, ML-193, and O-1602 (1, 5, 10, and 25 µM), LPS (10 ng/mL), ethanol (20 µL), and the solvent DMSO (0.1%), using an MTT assay (Figure 2). 

None of the tested compounds (KIT C, ML-193, O-1602) affected cell viability in concentrations of 10 and 25 µM. Neither LPS in the final concentration of 10 ng/mL nor the compounds showed any cytotoxic effects compared to untreated cells. A dose of 10 µM of ML-193 even increased cell viability/cell metabolism, measured as a reduction in MTT to formazan, compared to untreated cells. The solvent of the compounds, DMSO, did not alter cell viability on its own in the dose of 0.1%. As expected, 20 µL of ethanol significantly induced cell death as positive control [*F*(9, 24) = 65.15, *p* < 0.0001]. Based on those results, all compounds were used in concentrations up to 25 µM for the follow-up experiments. 

### 2.2. Effects of KIT C, ML-193, and O-1602 on TNF-α Expression and Release

TNF-α is a proinflammatory cytokine produced in cultured BV2 microglial cells after LPS stimulation [38]. Briefly, BV2 microglial cells were pretreated with KIT C, ML-193, and O-1602 (1, 5, 10, and 25 µM) for 30 min, followed by stimulation with LPS (10 ng/mL) for 4 h or 24 h, respectively.

LPS treatment significantly induced the expression (Figure 3A,C,E) and release (Figure 3B,D,F) of TNF-α compared to unstimulated cells. KIT C and ML-193 similarly showed a concentration-dependent decrease in LPS-induced TNF-α expression; both compounds reduced TNF-α release to baseline levels in the highest concentration of 25 µM. O-1602 showed a significant reduction in LPS-induced expression and release of TNF-α as well, but the effects on mRNA expression were smaller compared to KIT C and ML-193 and TNF-α synthesis was only inhibited in the concentrations of 10 and 25 µM, decreasing LPS-stimulated TNF-α release to baseline levels. 

### 2.3. Effects of KIT C, ML-193, and O-1602 on IL-6 Expression and Release

We next assessed the expression and release of IL-6 in LPS-treated BV2 microglial cells. Expression and release of IL-6 were strongly induced by LPS treatment. Again, KIT C and ML-193 showed a comparable concentration-dependent reduction in LPS-induced IL-6 mRNA expression to baseline levels of untreated cells (Figure 4A,C). O-1602 significantly inhibited LPS-induced IL-6 expression. However, the effect sizes were smaller when compared to KIT C or ML-193 (Figure 4E). All three compounds significantly reduced IL-6 release by approx. 50% in the highest concentration of 25 µM compared to the LPS positive control (Figure 4B,D,F). Therefore, the effects of KIT C and ML-193 on IL-6 mRNA expression were more distinctive than on IL-6 release, while O-1602 enfolded greater effects on IL-6 release than on IL-6 expression. 

### 2.4. Effects of KIT C, ML-193, and O-1602 on CCL2 Expression and Release

CCL2, as a small chemokine, attracts monocytes, T-memory cells, and dendritic cells (DCs) to sites of infection and inflammation [39]. LPS stimulation reliably induced CCL2 expression and release in BV2 microglial cells. KIT C and ML-193 concentration-dependently inhibited CCL2 mRNA expression, reaching basal CCL2 expression levels at concentrations of 10 µM (Figure 5A,C). O-1602 showed a concentration-dependent and significant reduction in CCL2 mRNA expression; however, at concentrations of 25 µM, O-1602 CCL2 expression was reduced to double the baseline expression rate only (Figure 5E). However, ML-193 and O-1602 showed a greater reduction in LPS-induced CCL2 release, with 25 µM of O-1602 even inhibiting CCL2 release under baseline concentrations (Figure 5D,F). KIT C significantly reduced CCL2 release starting at concentrations of 5 µM. However, no concentration-dependent effects were observed, and baseline levels were not reached at its highest concentration of 25 µM. 

### 2.5. Effects of KIT C, ML-193, and O-1602 on CCL3 Expression and Release

CCL3 is one of the most expressed chemokines during CNS inflammation. It is involved in different neurological and psychiatric disorders, such as multiple sclerosis or bipolar disorder [40]. CCL3 showed a higher basal expression and synthesis rate when compared to CCL2 (Figure 5 and Figure 6). CCL3 was significantly enhanced by LPS treatment (Figure 6). All compounds exerted a significant and concentration-dependent inhibition of LPS-induced CCL3 expression in BV2 microglial cells. The effects of KIT C and ML-193 were comparable, nearly reaching basal CCL3 expression at 25 µM, while O-1602 only reduced CCL3 expression by about 20% when compared to LPS-stimulated cells (Figure 6A,C,E). CCL3 release was strongly reduced under baseline synthesis by all three compounds, with KIT C showing the most pronounced effect (Figure 6B,D,F). 

### 2.6. Effects of KIT C, ML-193, and O-1602 on CXCL2 Expression and Release

CXCL2 attracts neutrophils to the site of inflammation [41]. Both CXCL2 expression and release in BV2 microglial cells were significantly upregulated by LPS treatment (Figure 7). All compounds showed significant inhibition of CXCL3 mRNA expression, with O-1602 showing the most concentration-dependent reduction starting at a concentration of 1 µM. ML-193 and KIT C showed a strong drop in CXCL2 expression levels between 5 and 10 µM, with both compounds strongly reducing CXCL2 in the two highest concentrations used (Figure 7A,C,E). However, the baseline mRNA expression of CXCL2 was not reached by any of the compounds at 25 µM. ML-193 and O-1602 showed significant and concentration-dependent inhibition of CXCL2 release, with O-1602 causing a reduction to 50% CXCL2 release of LPS-treated cells at 25 µM. KIT C significantly inhibited CXCL2 release, starting at concentrations of 5 µM. However, no concentration dependency was observed (Figure 7B,D,F).

### 2.7. Effects of KIT C, ML-193, and O-1602 on CXCL10 Expression and Release

CXCL10 promotes chemotaxis and modulates the inflammatory responses of cells [42]. CXCL10 expression and release were strongly induced by LPS stimulation of BV2 microglial cells. All compounds significantly and concentration-dependently inhibited LPS-induced CXCL10 mRNA expression, with KIT C and ML-193 starting with a significant inhibition at 1 µM. At 25 µM, both compounds showed a decreased expression of only 15% compared to LPS-treated cells. O-1602 showed smaller effects on CXCL10 mRNA expression starting at concentrations of 10 µM and reaching a maximal reduction of approx. 30% compared to LPS-treated cells (Figure 8A,C,E). CXCL10 release was significantly reduced by KIT C in all concentrations; a reduction of 40% compared to the LPS positive control was observed using 25 µM of KIT C (Figure 8B). ML-193 reduced CXCL10 release to below 50% of LPS-stimulated cells at 25 µM (Figure 8D). O-1602 showed comparable effects to KIT C at a concentration of 25 µM. However, lower doses of O-1602 showed less reduction of CXCL10 release compared to KIT C (Figure 8F).

### 2.8. Effects of KIT C, ML-193, and O-1602 on Phosphorylation of PKC (pan) (βII Ser660), p38 MAPK, ERK 1/2, and NF-κB

To elucidate the cellular signaling pathways targeted by the GPR55 antagonists and agonists used in this study, Western blots using phosphorylation-specific antibodies of kinases involved in neuroinflammation such as PKC, ERK-1/2, and p38 MAPK, and the NF-κB pathway were performed.

PKC is a family of protein serine/threonine kinases centrally involved in intracellular signal transduction. Several isoforms are known, which are divided into the conventional isoforms (PKCα, βI, βII, and γ), which require Ca^2+^, diacylglycerol, and phosphatidylserine. The novel isoforms (PKCδ, ε, η, and θ) need diacylglycerol and phosphatidylserine but are Ca^2+^-independent, while the atypical isoforms (PKCζ and λ/ι) are only dependent on phosphatidylserine [43]. All subfamilies participate in MAPK regulation, with Ras/Raf activation by PKC as an important way of activating MAPK [44,45]. The antibody used for PKC detection in the current study was phospho-PKC (pan) (βII Ser660) (Cell Signaling Technology, Frankfurt, Germany), detecting PKC α, βI, βII, δ, ε, η and θ isoforms when phosphorylated at a carboxy-terminal residue homologous to serine 660 of PKC βII (Cell Signaling Technology, Frankfurt, Germany). PKC isoforms seem to be involved in AD and mood disorders [46]. Furthermore, PKC might be involved in the antidepressant effects of some psychiatric drugs [47]. Therefore, the effects of the compounds on the phosphorylation and, thus, activation of PKC in LPS-stimulated BV2 microglia cells, were analyzed.

LPS significantly enhanced the basal phosphorylation of PKC (pan) (βII Ser660), with all compounds significantly inhibiting the phosphorylation at both highest concentrations used (Figure 9). Basal phosphorylation rates of PKC (pan) (βII Ser660) were observed after treatment with KIT C, ML-193, and O-1602 at 25 µM, with O-1602 even reducing phosphorylation of PKC under baseline levels (Figure 9C).

A growing body of research demonstrates the vital role of the MAPK signaling pathway in brain inflammation and neurodegeneration [48,49]. Therefore, the effects of the compounds on MAPK signaling pathways were studied, focusing on the phosphorylation and consecutive activation of ERK 1/2 and p38 MAPK in LPS-stimulated BV2 microglia cells.

The phosphorylation of ERK 1/2 was induced by treatment with LPS, but in the experiments for KIT C, a higher basal expression or less potent induction of ERK 1/2 phosphorylation was observed compared to the results of ML-193 and O-1602. ML-193 and O-1602 showed a concentration-dependent inhibition of ERK 1/2 phosphorylation, reaching significance at both of the highest concentrations (Figure 10B,C). For KIT C, no concentration-dependent effect on LPS-induced ERK 1/2 phosphorylation was observed. However, all used concentrations tended to decrease ERK 1/2 phosphorylation, reaching significance at the highest dose of 25 µM (Figure 10A).

The phosphorylation of p38 MAPK was significantly induced by LPS stimulation in BV2 microglial cells. KIT C inhibited p38 MAPK phosphorylation in its highest concentration of 25 µM, and ML-193 showed a significant reduction in p38 MAPK phosphorylation starting at 10 µM. ML-193 showed a more potent reduction than KIT C (Figure 11A,B). O-1602 did not significantly alter p38 MAPK phosphorylation (Figure 11C). However, a tendency to increase p38 MAPK phosphorylation at both of the highest concentrations was observed.

NF-κB signaling is involved in various cellular processes; its most important role is mediating inflammatory responses [50]. Therefore, we studied the effects of KIT C, ML-193, and O-1602 on the phosphorylation of NF-κBp65, as shown in Figure 11. Treatment of BV2 microglial cells reliably induced NF-κBp65 phosphorylation. ML-193 significantly and concentration-dependently inhibited the LPS-induced phosphorylation of NF-κBp65 (Figure 12B). In contrast, KIT C and O-1602 did not alter the phosphorylation of NF-κBp65 significantly. O-1602 tended to increase the phosphorylation of NF-κBp65 compared to LPS-treated positive control (Figure 12C), while KIT C did not show any tendency at any concentration (Figure 12A).

## 3. Discussion

Neuroinflammation is considered to be a vital factor in the process of neurodegeneration [51]. Microglia, as the resident innate immune cells of the brain, are a key component of neuroinflammatory responses [52]. The understanding of possible molecular mechanisms leading to neurological and psychiatric disorders is increasing, resulting in a growing interest in developing drugs targeting microglia and controlling neuroinflammatory processes [53]. The current study investigates the anti-neuroinflammatory effects of the coumarin derivate KIT C, which are compared to the effects of the commercial GPR55 agonist O-1602 and the GPR55 antagonist ML-193 in LPS-stimulated BV2 microglia cells. All compounds showed significant inhibition of IL-6 and TNF-α expression and release, but the effects of KIT C and ML-193 exceeded the effects of O-1602 for cytokine gene expression. Figure 13 summarizes the observed effects of LPS and KIT C on the investigated cytokines, chemokines, and enzymes in the BV2 microglial cells.

Furthermore, all compounds significantly inhibited the expression and release of the chemokines CCL2, CCL3, CXCL2, and CXCL10. Again, especially for the gene expression of the examined chemokines, KIT C and ML-193 showed more potent inhibitory effects than O-1602, which was partially ameliorated when chemokine protein release was compared between the three compounds. Finally, the tested compounds reduced the phosphorylation of PKC (pan) (βII Ser660) and ERK 1/2. The results for p38 MAPK and NF-κBp65 differed between the GPR55 ligands.

G-protein-coupled receptors (GPCRs) have been identified as an important class of receptors in the pathogenesis of psychiatric disorders, and most neurotransmitters enfold their physiological effects by different types of GPCRs [54]. GPR55, a receptor considered part of the endocannabinoid system, has gained much attention in recent years because of its promising role in the modulation of neuroinflammation and the treatment of inflammatory-mediated neurological and psychiatric disorders [35]. Coumarin-based compounds act as inverse agonists at the GPR55 when substituted on C3 and C8 [34]. Previous studies found promising GPR55-dependent anti-inflammatory and anti-oxidative properties of KIT C in human SK-N-SH neuroblastoma cells and primary mouse microglia [32]. The anti-neuroinflammatory effects of other coumarin derivates have been demonstrated in primary rat microglia, potentially inhibiting LPS-induced PGE_2_ release [27]. Inhibition of PGE_2_ release was associated with inhibition of COX-2 synthesis. However, no significant effects of the coumarin derivatives on MAPKs were observed [19,27,32].

As a GPR55 antagonist with inverse agonistic properties, KIT C showed similar anti-inflammatory effects on the expression and release of different cytokines and chemokines compared to the commercial GPR55 antagonist ML-193. In a previous study, KIT C showed comparable effects to ML-193 regarding inhibition of IL-1β-induced PGE_2_ release in SK-N-SH cells, while O-1602 did not significantly reduce PGE_2_ release [32]. The same results were observed in LPS-stimulated primary rat microglia, with ML-193 significantly inhibiting PGE_2_ release while O-1602 did not modulate PGE_2_ synthesis [19]. However, in the current study, O-1602 reduced chemokine and cytokine expression and release, and chemokine release was reduced in comparison to ML-193. This stands in direct contrast to our previous findings in IL-1β-stimulated SK-N-SH cells, where ML-193 inhibited IL-6 release, while O-1602 did not alter IL-6 release [32]. The contrary results and similar behavior of GPR55 agonists and antagonists in the current study compared to our previous studies might be explained by the use of different cell types, since neurons and microglial cells may vary in receptor density, species-specific effects due to differences in the molecular structure of the GPR55, or different associated GPCR transducers. Furthermore, the stimulation of SK-N-SH cells with IL-1β [32] activates different pathways than LPS stimulation; therefore, the different stimulants used might be another explanation for differences in the observed effects between recent studies and our previous study.

Furthermore, alternative signaling pathways, in terms of biased agonisms, might be activated in different cells dependent on alternated G-protein coupling to the GPR55, as discussed previously [32]. Receptor-independent pathways might explain the similar effects of GPR55 agonists and antagonists on the expression and release of the investigated cytokines and chemokines. Future studies using GPR55 knock-out BV2 cells could help to differentiate receptor-dependent and receptor-independent effects of the tested compounds.

Another possible explanation for the observed differences between the current study and our previous studies in SK-N-SH cells is using different stimuli for inflammatory responses. IL-1β activates the MAPK/NF-κB pathway via heterodimerization of the IL-1 receptor 1 (IL-1R1) and IL-1R3, mediating acute and chronic inflammatory processes by regulating gene expression [55]. The most relevant receptor for LPS signaling is the toll-like receptor 4 (TLR4), resulting in NF-κB activation as well [56]. Similar cell responses to IL-1β and LPS are explained by both receptors’ shared Toll/IL-1R (TIR) domain. However, different adapter proteins, such as MD-2 forming an extracellular LPS/MD-2 complex for TLR4 binding, modulate MAPK and NF-κB activation depending on the specific pathogen-associated molecular pattern (PAMP) cells are confronted with [56]. Furthermore, the BV2 cell line’s immortalization process might alter cellular responses compared to primary cell cultures. Different molecular signatures have been revealed between BV2 cells, other commercial microglial cell lines, and primary microglia, possibly resulting in different inflammatory signaling [57].

We further investigated pathways possibly involved in the synthesis of cytokine and chemokines to understand the molecular mechanisms of the effects of KIT C on inflammatory processes. LPS exerts proinflammatory responses via MAPK/NF-κB signaling [56]. In line with previous studies, LPS induced MAPK and NF-κBp65 phosphorylation in LPS-stimulated BV2 cells.

Although all tested compounds in the current study showed potent cytokine and chemokine release inhibition, only PKC (pan) (βII Ser660) and ERK 1/2 phosphorylation were reduced by all compounds. Several studies suggest that the PKC pathway is a key modulator of inflammatory responses [58]. Furthermore, the PKC/Ca^2+^ pathway plays an important role in the NF-κB pathway activation induced by different stimuli [59,60]. At least 11 isoforms of PKC are known, but the current study only evaluated the effects of the compounds on the PKC βII Ser660 phosphorylation site detecting PKC α, βI, βII, δ, ε, η and θ isoforms when they phosphorylated at a carboxy-terminal residue homologous to serine 660 of PKC βII and did not distinguish between the different isoforms. Future studies should therefore investigate the effects of the compounds on distinct different PKC isoforms and phosphorylation sites and the consequences for the downstream signaling. p38 MAPK phosphorylation was inhibited by KIT C and ML-193 in the highest concentration of 25 µM, but significant effects on cytokine and chemokine expression and release were also observed in lower concentrations. However, inhibition of p38 MAPK reduced CXCL10 synthesis in a poly(I:C)-stimulated human bronchial epithelial cell line and human natural killer cells [61], underlining the relevance of MAPK pathways in chemokine regulation. Since MAPK signaling pathways may contribute to the pathogenesis of AD by regulation of neuronal apoptosis [62], phosphorylation of amyloid precursor protein (APP) and tau-protein [63], and other neuropsychiatric disorders, targeting MAPK pathways might offer new therapeutic approaches.

NF-κB regulates gene expression of different inflammatory mediators, including cytokines and chemokines [64]. In the current study, only ML-193 significantly altered NF-κB phosphorylation. Therefore, the tested compounds may act via additional steps in the signaling pathways, contributing to the observed anti-inflammatory effects. NF-κB is complexly regulated by Keap1 [65,66] and IκBα [67], which were not investigated in the current study. Furthermore, PLC and PKC trigger intracellular Ca^2+^ release, acting as second messenger and contributing to differentiated cell responses [68]. However, since NF-κB modulates Beta-Secretase 1 (BACE1) expression, closely related to AD [69], and inhibition of NF-κB prevents neuronal apoptosis in different neurodegenerative disorders [70], targeting NF-κB signaling again opens promising therapeutical options. Further research is necessary to fully understand the GPR55-dependent modulation of inflammatory effects and the possible indirect effects on the NF-κB pathway via IκB or Keap1. This could help boost the efficacy of treating inflammatory-related disorders with anti-inflammatory substances.

Altered levels of cytokines and chemokines are reported in neurological and psychiatric disorders [11,71,72], and for depression, psychotherapy, and antidepressants reduced levels of IL-6 [73]. Based on the inflammatory hypothesis of psychiatric disorders, such as depression [74], anti-inflammatory treatment might offer new therapeutical approaches for those disorders beyond the modulation of neurotransmitter levels. The potential of GPR55 agonists and antagonists is the subject of research in the modulation of inflammation and behavioral aberrations in animal disease models.

In 5xFAD mice, a genetic AD animal model, hippocampal GPR55 expression was increased and was associated with memory impairment and anxiety-like behavior [75]. In a streptozotocin (STZ)-induced AD mouse model, intracerebroventricular (i.c.v.) administration of O-1602 ameliorated cognitive dysfunction. O-1602 normalized the STZ-induced reduction of GPR55 and acetylcholinesterase (AChE) and upregulation of BACE1 and beta-amyloid [76]. Similarly, in an LPS-induced murine AD model, O-1602 reversed LPS-dependent downregulation of GPR55, cognitive impairment, and neuronal apoptosis. O-1602 prevented LPS-induced hippocampal release of proinflammatory cytokines and oxidative stress via NF-κB signaling [77]. Transgenic CCL2 overexpression promoted glial activation with increased IL-6 levels and tau pathology acceleration in a mouse tau pathology model [78]. Furthermore, chronically increased levels of CCL3 are described in AD patients [79]. CXCL10 levels in cerebrospinal fluid (CSF) were higher in patients with AD and mild cognitive impairment, possibly based on caspase-dependent CXCL10-induced apoptosis of fetal neurons [42]. Additionally, increased levels of CXCL10 correlated with markers of intrathecal inflammation were found in CSF of patients with multiple sclerosis (MS), while CCL2 was suppressed [80]. CXCL2 is elevated in Alzheimer’s disease [81] and amyotrophic lateral sclerosis (ALS) [82]. Therefore, inhibition of chemokine synthesis might be a promising therapeutic approach in those neurological disorders.

A 6-hydroxydopamine-induced rat PD model showed reduced sensorimotor and motor functions, which were prevented by intrastriatal administration of the GPR55 agonist lysophosphatidylinositol (LPI), as well as the antagonist ML-193, suggesting a modulatory role of GPR55 in PD [83]. GPR55 knock-out mice expressed anxiety and motor activity levels comparable to those of wild-type mice while showing impaired thermal sensitivity and motor coordination [84]. Abnormal cannabidiol, another GPR55 agonist, prevented motor impairment in a 1-methyl-4-phenyl-1,2,3,6-tetrahydropyridine (MPTP) or haloperidol-induced murine PD model. The positive effects of abnormal cannabidiol were abolished by treatment with GPR55 antagonists in the haloperidol PD model, suggesting the effects are only mediated by GPR55 activation [85]. Since inflammatory processes contribute to PD pathogenesis [71] and the positive effects of GPR55 activation have already been shown, reducing cytokine and chemokine levels via GPR55 activation might be a promising approach in novel PD therapies.

Furthermore, GPR55 is also investigated for its role in depression. In Wistar rats, the GPR55 agonist O-1602 promoted anxiolytic-like behavior, while the antagonist ML-193 elicited anxiety-like behavior [28]. Furthermore, O-1602 reversed symptoms of depression in rats subjected to corticosterone treatment and normalized altered hippocampal inflammatory markers [30]. Chronic social defeat stress (CSDS) induced depression- and anxiety-like behavior in mice was associated with a reduced hippocampal GPR55 expression. Electroacupuncture or GPR55 activation by O-1602 both ameliorated CSDS-induced depression- and anxiety-like behavior, and the GPR55 antagonist CID16020046 reversed the positive effects of electroacupuncture [86]. Those studies showed that GPR55 activation leads to reduced depressive-like behavior that might be dependent on altered inflammatory processes, suggesting new therapeutical approaches toward psychiatric disorders.

As an inverse agonist of GPR55, KIT C exerts anti-inflammatory and anti-oxidative properties [32,35]. The current study further supports the anti-inflammatory effects, showing potent inhibition of cytokine and chemokine expression and release in LPS-stimulated BV2 cells. GPR55 selective coumarin derivatives should therefore be further investigated for their potential use in inflammation-related neurological and psychiatric disorders.

## 4. Materials and Methods

### 4.1. Chemicals

KIT C ((8-isopropyl-3,5-dimethyl-2H-chromen-2-one)) was synthesized as described previously [34,35] by the Karlsruhe Institute for Technology (KIT), Institute of Organic Chemistry, dissolved in DMSO (cat. no.: 8.02912.1 000, Merck KGaA, Darmstadt, Germany), and used in final concentrations of 1, 5, 10, and 25 µM. The commercially available GPR55 agonist O-1602 and antagonist ML-193 (cat. no.: 10006803 and 15184, both from Cayman Chemicals, Ann Arbor, MI, USA, distributed by BioMol, Hamburg, Germany) were dissolved in DMSO and used in final concentrations of 1, 5, 10, and 25 µM. A total of 5 mg/mL lipopolysaccharide (LPS) from Salmonella typhimurium (cat. no.: L6143, Sigma Aldrich, Deissenhofen, Germany) was dissolved in PBS as stock and diluted with distilled water for a final concentration of 10 ng/mL in BV2 microglial cultures.

### 4.2. BV2 Microglial Cell Culture

Immortalized BV2 microglial cells (kindly provided by Prof. Langmann, Department of Opthalmology, University of Cologne, Cologne, Germany), were cultured in 1x RPMI 1640 Medium containing 10% fetal calf serum (cat. no.: FCS.SAF.0.500, FCS, Bio and SELL GmbH, Feucht/Nürnberg, Germany), 2 mM L-glutamine, and 1% penicillin/streptomycin (cat. no.: 25030081 and 11548876, all cell culture solutions obtained by Gibco, Thermo Fisher Scientific, Bonn, Germany). The culture environment was 5% CO_2_ at 37 °C in a humidified atmosphere. Cells were grown to approx. 90% confluency, regularly passaged by trypsinization and re-seeded to 6-, 12-, 24-, or 96-well plates or new cell culture flasks, respectively. The medium was replaced by fresh medium 1 h before the experiments. BV2 cells were cultured and used to a maximum of 30 passages.

### 4.3. Cell Viability Assay

An MTT assay (cat. no.: M-2128, Sigma-Aldrich GmbH, Taufkirchen, Germany) was used for measuring the viability of BV2 after treatment with KIT C, O-1602, or ML-193 (1, 5, 10, and 25 µM). Based on the reduction of a yellow tetrazolium salt (3-(4,5-dimethylthiazol-2-yl)-2,5-diphenyltetrazolium bromide or MTT) to purple formazan, the assay determines the number of metabolically active cells. Therefore, conclusions about viable cells in the culture can be drawn. Briefly, cells were cultured in 96-well plates at a density of approximately 25 × 10^3^ cells/well for 24 h. The medium was changed and after at least 1 h, cells were pretreated with different concentrations of the compounds for 30 min and then incubated with LPS (10 ng/mL) for the next 20 h. Untreated, DMSO, and LPS-treated cells served as controls. As a positive control, 20 µL ethanol (approximately 20% end conc.) was used to induce cell death. Next, 20 µL of the MTT-solution (working concentration 5 mg/mL) were added to all wells and incubated for another 4 h at 37 °C. Afterwards, the medium was removed and replaced by 200 µL of DMSO. Colorimetric reaction was measured using MRXe Microplate reader (Dynex Technologies, Denkerdorf, Germany) with photometric extinction at 570 nm wavelength and a reference wavelength of 630 nm.

### 4.4. Determination of Cytokine and Chemokine Release

Commercially available ELISA kits were used following the manufacturer’s protocol for the determination of murine TNF-α, IL-6, CCL2, CCL3, CXCL2, and CXCL10 concentrations (cat. no.: DY410, DY406, DY479, DY450, DY452 and DY466, bio-techne/R&D Systems, Wiesbaden, Germany). Briefly, cultured BV2 microglial cells were left untreated or incubated with KIT C, ML-193, or O-1602 (1, 5, 10, and 25 μM) for 30 min. Afterwards, LPS (10 ng/mL) was added to the appropriate wells, and cells were incubated for 24 h. Supernatants were collected and stored at −80 °C after centrifugation at 1000× *g* for 2 min at 4 °C. ELISA plates (Nunc MaxiSorpTM; Thermo Fisher Scientific, Bonn, Germany) were coated with the respective capture antibody overnight. The next day, samples were added, followed by the addition of the appropriate determination antibody after removing supernatants and washing the plate. The amount of bound determination antibody was quantified using an HRP-dependent colorimetric reaction. The absorbance of the wells was read at 450 nm using the MRXe Microplate reader. The results were normalized to LPS and presented as a percentage of change in cytokine and chemokine levels.

### 4.5. RNA Isolation and Quantitative PCR

Gene expression of the cytokines and chemokines IL-6, TNF-α, CCL2, CCL3, CXCL2, and CXCL10 was measured using quantitative real-time PCR (qPCR). Cultured BV2 microglial cells were left untreated or pretreated with KIT C, ML-193, or O-1602 (1, 5, 10, and 25 µM) for 30 min, followed by stimulation with LPS (10 ng/mL) for 4 h. Total RNA was extracted using the GeneMATRIX Universal RNA Purification Kit (cat. no.: E3598, Roboklon GmbH, Berlin, Germany), according to the manufacturer’s protocol. Then, cDNA was reverse transcribed from 500 ng of total RNA in a 30 μL total reaction volume with initial denaturation at 70 °C (10 min; with random primers, biomers.net GmbH, Ulm, Germany) with a following amplification cycle after addition of master mix (cDNA master mix: M-MLV reverse transcriptase (cat. no.: M-1708), RNAsin® Ribonuclease Inhibitor (cat. no.: N-251B), RT-Buffer M-MLV 5x (cat. no.: M531A); Promega GmbH, Mannheim, Germany). qPCR amplification was carried out by the CFX384 real-time PCR detection system (Bio-Rad Laboratories GmbH, Feldkirchen, Germany) using SYBR green as a fluorescent dye (cat. no.: 4472942, Applied Biosystems SYBR® Select Master Mix for CFX, Thermo Fisher Scientific, Bonn, Germany). Glyceraldehyde 3-phosphate dehydrogenase (GAPDH) was an internal control for sample normalization. The primer sequences were as follows, GAPDH: Forward (Fwd): 5′-TGGGAAGCTGGTCATCAAC-3′/Reverse (Rev): 5′-GCATCACCCCATTTGATGTT-3′, TNF-α: Fwd: 5′-CCCACGTCGTAGCAAACCACCA-3′/Rev: 5′-CCATTGGCCAGGAGGGCGTTG-3′, IL-6: Fwd: 5′-AGTTGCCTTCTTGGGACTGA-3′/Rev: 5′-TTCTGCAAGTGCATCATCGT-3′, CCL2: Fwd: 5′-TGATCCCAATGAGTAGGCTGG-3′/Rev: 5′-ACCTCTCTCTTGAGCTTGGTG-3′, CCL3: Fwd: 5′-TATTTTGAAACCAGCAGCCTTT-3′/Rev: 5′-ATTCTTGGACCCAGGTCTCTTT-3′, CXCL2: Fwd: 5′-CCCTCAACGGAAGAACCAAAG-3′/Rev: 5′-GAGGCACATCAGGTACGATCCA-3′, and CXCL10: Fwd: 5′-CAGTGGATGGCTAGTCCTAATTG-3′/Rev: 5′-ACTCAGACCAGCCCTTAAAGAAT-3′. Primers were designed using Universal ProbeLibrary Assay Design Center (Roche Diagnostics, Mannheim, Germany) or sequence was designed by hand based on the target gene with orientation to existing literature, and were obtained by biomers.net GmbH (Ulm, Germany).

### 4.6. Immunoblotting

BV2 microglial cells were left untreated or pretreated with KIT C, ML-193, or O-1602 (1, 5, 10, and 25 μM) for 30 min and then stimulated with LPS (10 ng/mL) for 30 min. Afterward, cells were collected after being washed with cold PBS and lysed in lysis buffer (42 mM Tris–HCl, 1.3% sodium dodecyl sulfate, 6.5% glycerin, 100 μM sodium orthovanadate, and 2% phosphatase and protease inhibitors). Protein concentrations of the samples were measured using the bicinchoninic acid (BCA) protein assay kit (cat. no.: 23225, Fisher Scientific GmbH, Schwerte, Germany) according to the manufacturer’s instructions. For Western blotting, 20 μg of total protein in each sample was subjected to 10% sodium dodecyl sulfate-polyacrylamide gel electrophoresis (SDS-PAGE) under reducing conditions. Afterwards, proteins were transferred onto 0.45 µm polyvinylidene fluoride (PVDF) membrane (cat. no.: 88518, Merck Millipore, Darmstadt, Germany) via semi-dry blotting for 1 h. Blocking was performed using Roti-bock (cat. no.: A151, Roth, Karlsruhe, Germany) for 1 h. Membranes were then incubated overnight with the respective primary antibodies. Primary antibodies were rabbit anti-phospho-PKC (pan) (βII Ser660) (cat. no.: 9371S, 1:1000; Cell Signaling Technology, Frankfurt, Germany), rabbit anti-ERK1/2 (cat. no.: 9101S, 1:1000; Cell Signaling Technology, Frankfurt, Germany), rabbit anti-p38 MAPK (cat. no.: 9211S, 1:1000; Cell Signaling Technology), rabbit anti-phospho-NF-kappaB-p65 (cat. No.: 3031S, 1:1000; Cell Signaling Technology, Frankfurt, Germany), and mouse anti-Vinculin (cat. no.: V9264, 1:20,000, Sigma-Aldrich GmbH, Taufkirchen, Germany). Protein-bound antibodies were detected with horseradish peroxidase-coupled goat anti-rabbit IgG (cat. no.: NA9340V, Amersham Biosciences GmbH, Freiburg, Germany, 1:10,000 dilution), mouse anti-rabbit IgG (cat. no.: NA934, Amersham Biosciences GmbH, Freiburg, Germany, 1:25,000 dilution), or sheep anti-mouse IgG (cat. no.: NA931V, 1:10,000 dilution; Amersham Biosciences GmbH, Freiburg, Germany) using enhanced chemiluminescence (ECL) reagents (cat. no.: 541015, Biozym, Hessisch Oldendorf, Germany). Densitometric analysis was performed using ImageJ software 1.48t (NIH, Bethesda, MD, USA).

### 4.7. Statistical Analysis

Raw values were converted to percentages considering LPS (10 ng/mL) or the appropriate positive control, such as untreated cells for MTT assay, as 100%. No outlier statistics were performed, and no values were excluded from further statistics. Data are represented as mean ± SD of at least three independent experiments. The statistical comparisons were performed using one-way ANOVA with Dunnett’s post hoc test (Prism 8, GraphPad Software Inc., San Diego, CA, USA). The significance level was set at * *p* < 0.05, ** *p* < 0.01, *** *p* < 0.001 and **** *p* < 0.0001 and is indicated in the figures.

## 5. Conclusions

Numerous neurological and psychiatric disorders are associated with altered levels of inflammatory markers. However, effective pharmacological treatment of those disorders remains a challenge. Novel targets for pharmacological treatment might offer more effective therapy options with fewer side effects based on the inflammatory hypothesis of neurological and psychiatric disorders. GPR55, as part of the endocannabinoid system, has gained interest in relation to the pathogenesis of various neurological and psychiatric disorders, and different studies demonstrate its relevance for disorders such as AD, PD, and depression. The coumarin derivate KIT C, a GPR55 antagonist with inverse agonistic activities, known for its anti-inflammatory and -oxidative effects, potently inhibited cytokine and chemokine expression and release in LPS-stimulated BV2 cells. The effects were comparable to the commercial GPR55 antagonist ML-193. Therefore, KIT C should be further investigated as a promising compound for inflammatory disorders.

## Figures and Tables

**Figure 1 pharmaceuticals-17-00674-f001:**
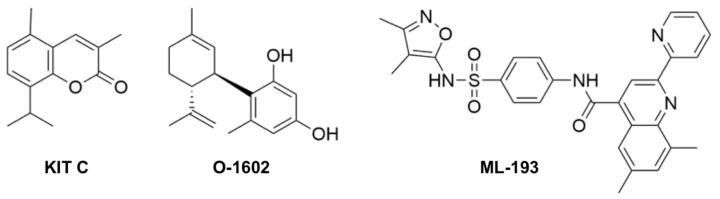
Molecular structure of the coumarin derivative KIT C [35], and the commercial GPR55 ligands O-1602 and ML-193.

**Figure 2 pharmaceuticals-17-00674-f002:**
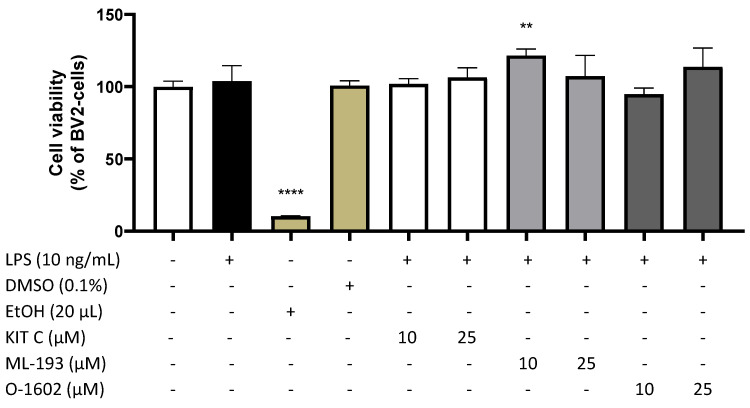
Effects of KIT C (white bars, except negative control), ML-193 (light grey), and O-1602 (dark grey) on cell viability in LPS-stimulated BV2 microglial cells. Cell viability was measured after 24 h of treatment by color change due to MTT oxidation. Values are presented as the mean ± SD of three independent experiments. Statistical analysis was performed using one-way ANOVA with Dunnett’s post hoc test with ** *p* < 0.01 and **** *p* < 0.0001 compared to untreated cells.

**Figure 3 pharmaceuticals-17-00674-f003:**
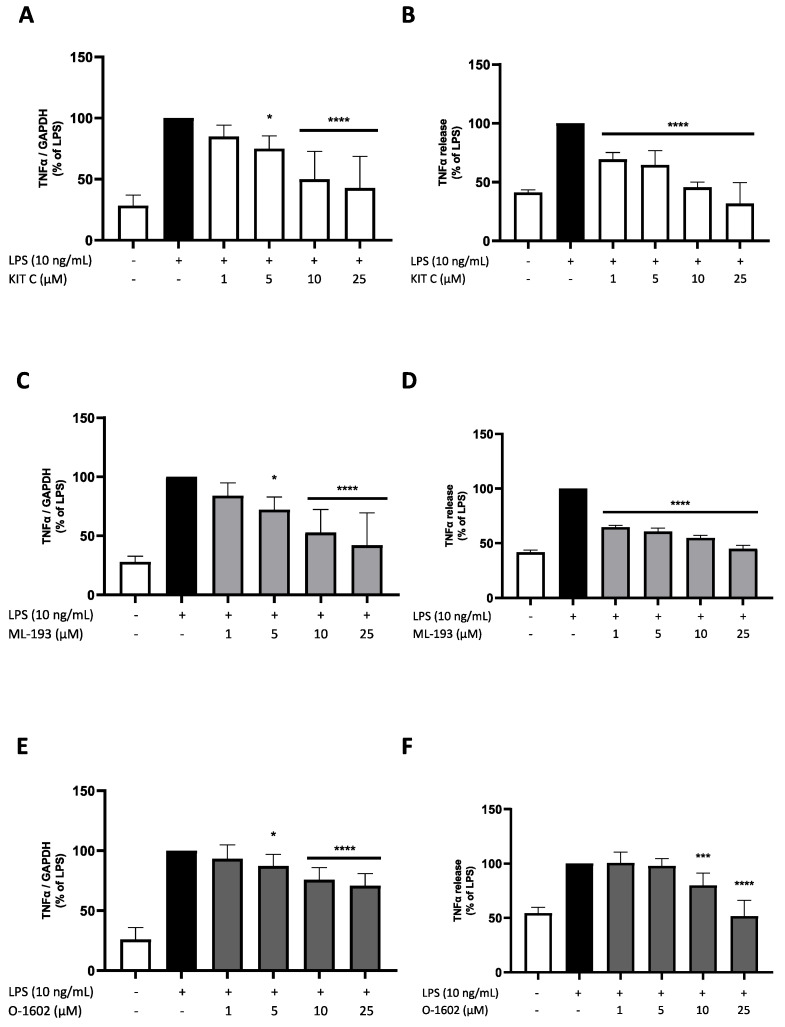
Effects of KIT C (**A**,**B**), ML-193 (**C**,**D**), and O-1602 (**E**,**F**) on the expression (**A**,**C**,**E**) and release (**B**,**D**,**F**) of TNFα in LPS-stimulated BV2 cells. Cells were stimulated as described in the Materials and Methods section (4 h for qPCR, 24 h for ELISA). Values are presented as the mean ± SD of at least three independent experiments (N = 3 (**D**), N = 6 (**A**–**C**), N = 9 (**E**,**F**)). Statistical analysis was performed using one-way ANOVA with Dunnett’s post hoc tests with * *p* < 0.05, *** *p* < 0.001, and **** *p* < 0.0001 compared to LPS-stimulated cells.

**Figure 4 pharmaceuticals-17-00674-f004:**
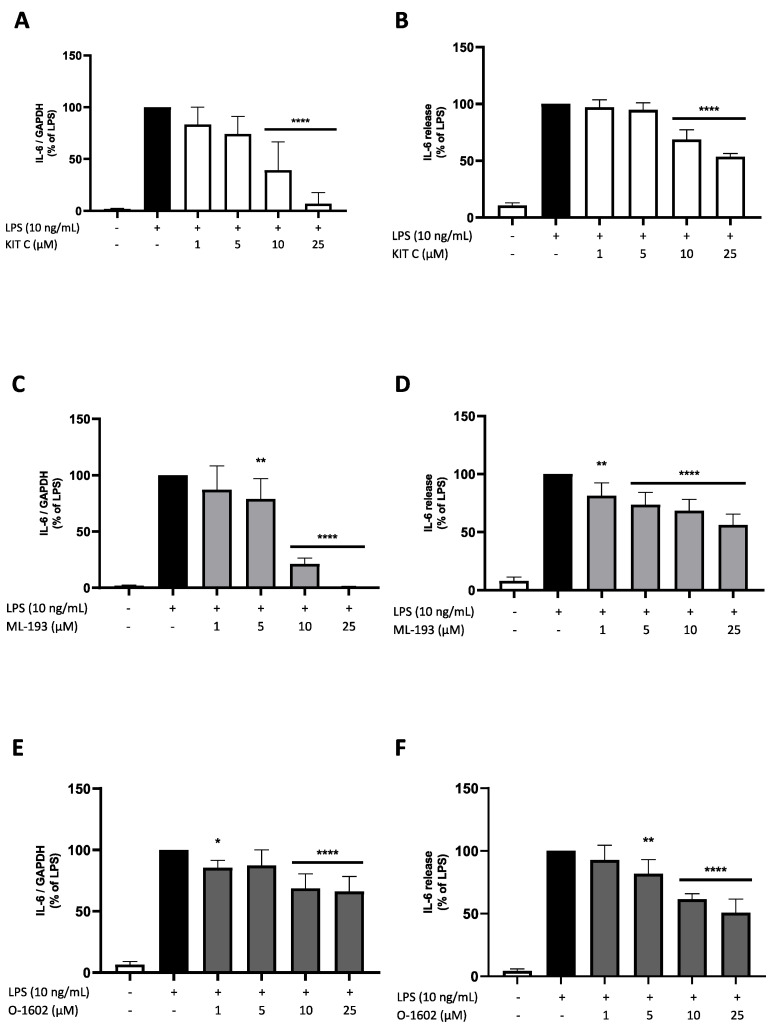
Effects of KIT C (**A**,**B**), ML-193 (**C**,**D**), and O-1602 (**E**,**F**) on the expression (**A**,**C**,**E**) and release (**B**,**D**,**F**) of IL-6 in LPS-stimulated BV2 cells. Cells were stimulated as described in the Materials and Methods section (4 h for qPCR, 24 h for ELISA). Values are presented as the mean ± SD of at least three independent experiments (N = 3 (**B**), N = 5 (**A**), N = 6 (**C**,**D**,**F**), N = 7 (**E**)). Statistical analysis was performed using one-way ANOVA with Dunnett’s post hoc tests with * *p* < 0.05, ** *p* < 0.01, and **** *p* < 0.0001 compared to LPS-stimulated cells.

**Figure 5 pharmaceuticals-17-00674-f005:**
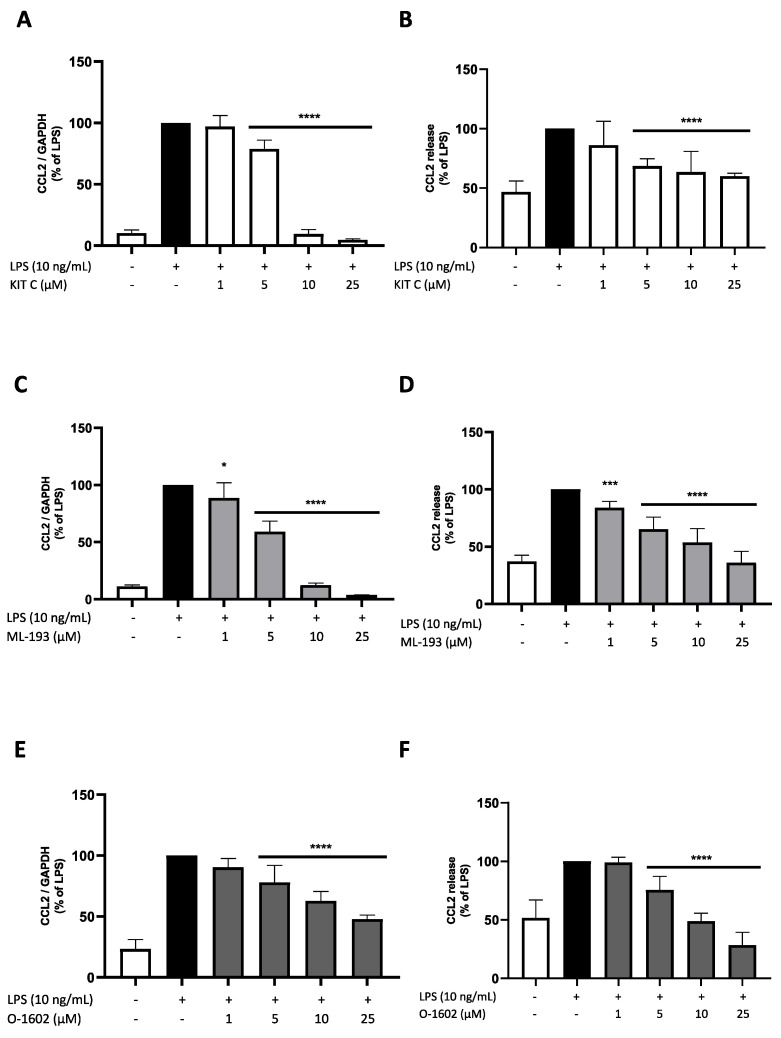
Effects of KIT C (**A**,**B**), ML-193 (**C**,**D**) and O-1602 (**E**,**F**) on the expression (**A**,**C**,**E**) and release (**B**,**D**,**F**) of CCL2 in LPS-stimulated BV2 cells. Cells were stimulated as described in the Materials and Methods section (4 h for qPCR, 24 h for ELISA). Values are presented as the mean ± SD of at least three independent experiments (N = 6 (**A**,**C**), N = 7 (**E**), N = 9 (**B**), N = 10 (**D**), N = 11 (**F**)). Statistical analysis was performed using one-way ANOVA with Dunnett’s post hoc tests with * *p* < 0.05, *** *p* < 0.001 and **** *p* < 0.0001 compared to LPS-stimulated cells.

**Figure 6 pharmaceuticals-17-00674-f006:**
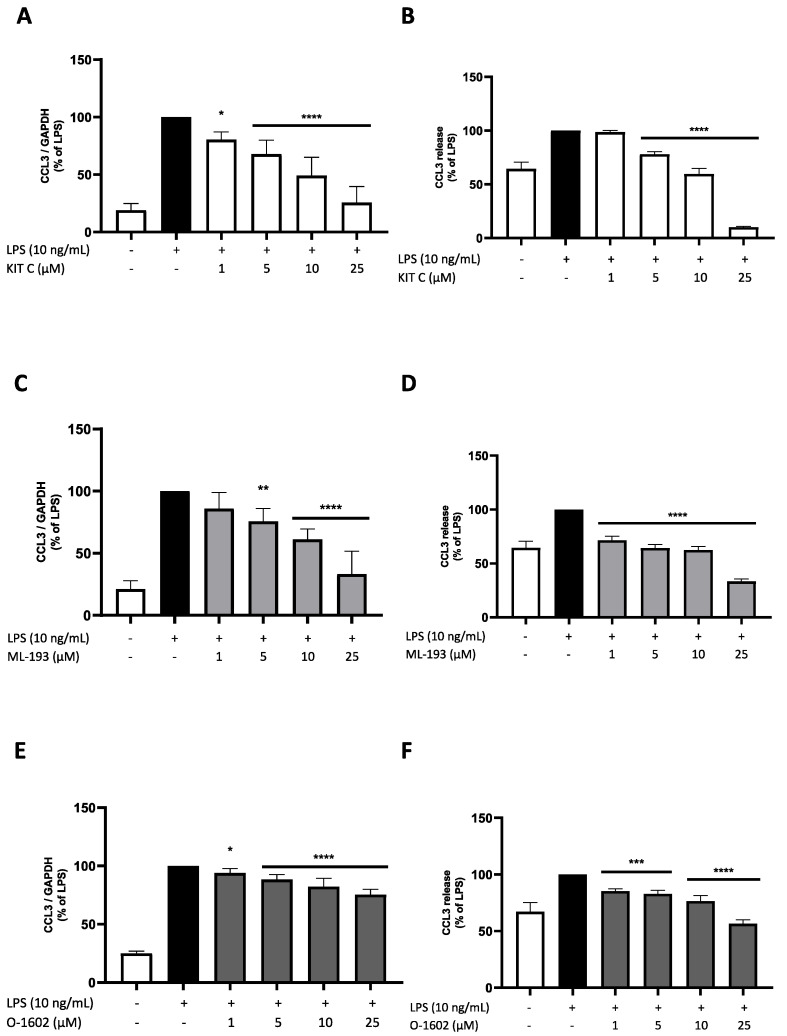
Effects of KIT C (**A**,**B**), ML-193 (**C**,**D**) and O-1602 (**E**,**F**) on the expression (**A**,**C**,**E**) and release (**B**,**D**,**F**) of CCL3 in LPS-stimulated BV2 cells. Cells were stimulated as described in the Materials and Methods section (4 h for qPCR, 24 h for ELISA). Values are presented as the mean ± SD of at least three independent experiments (N = 3 (**B**,**D**), N = 4 (**F**), N = 6 (**A**,**C**), N = 7 (**E**)). Statistical analysis was performed using one-way ANOVA with Dunnett’s post hoc tests with * *p*< 0.05, ** *p* < 0.01, *** *p* < 0.001, and **** *p* < 0.0001 compared to LPS-stimulated cells.

**Figure 7 pharmaceuticals-17-00674-f007:**
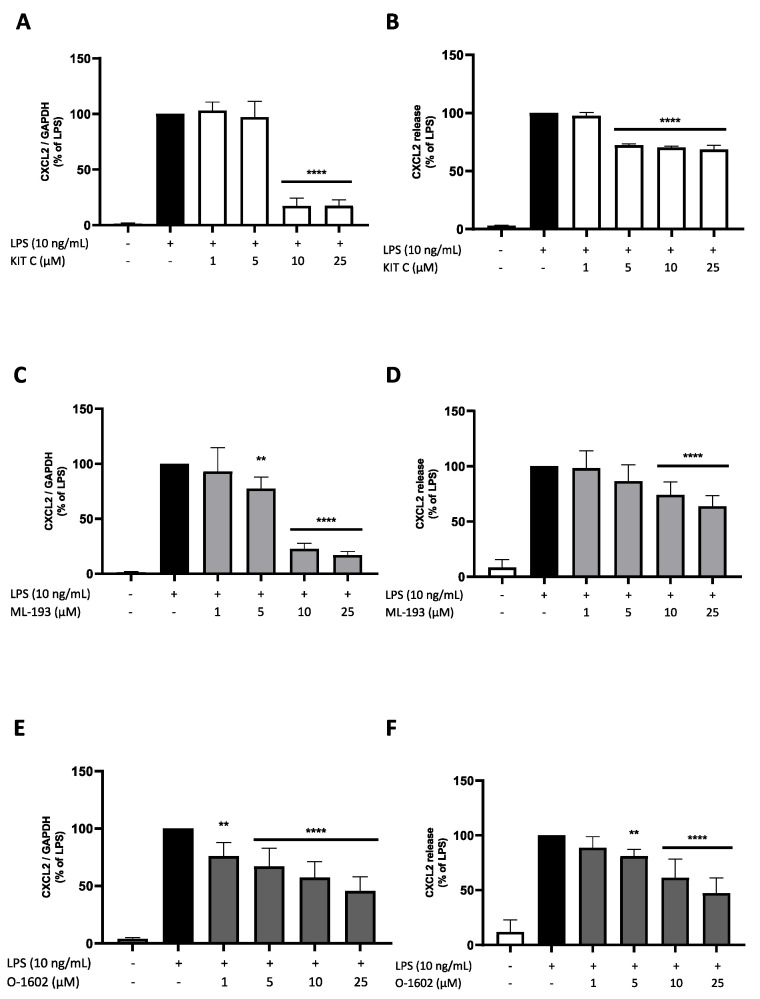
Effects of KIT C (**A**,**B**), ML-193 (**C**,**D**), and O-1602 (**E**,**F**) on the expression (**A**,**C**,**E**) and release (**B**,**D**,**F**) of CXCL2 in LPS-stimulated BV2 cells. Cells were stimulated as described in the Materials and Methods section (4 h for qPCR, 24 h for ELISA). Values are presented as the mean ± SD of at least three independent experiments (N = 3 (**B**), N = 6 (**A**,**C**), N = 7 (**E**), N = 8 (**F**), N = 9 (**D**)). Statistical analysis was performed using one-way ANOVA with Dunnett’s post hoc tests with ** *p*< 0.01 and **** *p* < 0.0001 compared to LPS-stimulated cells.

**Figure 8 pharmaceuticals-17-00674-f008:**
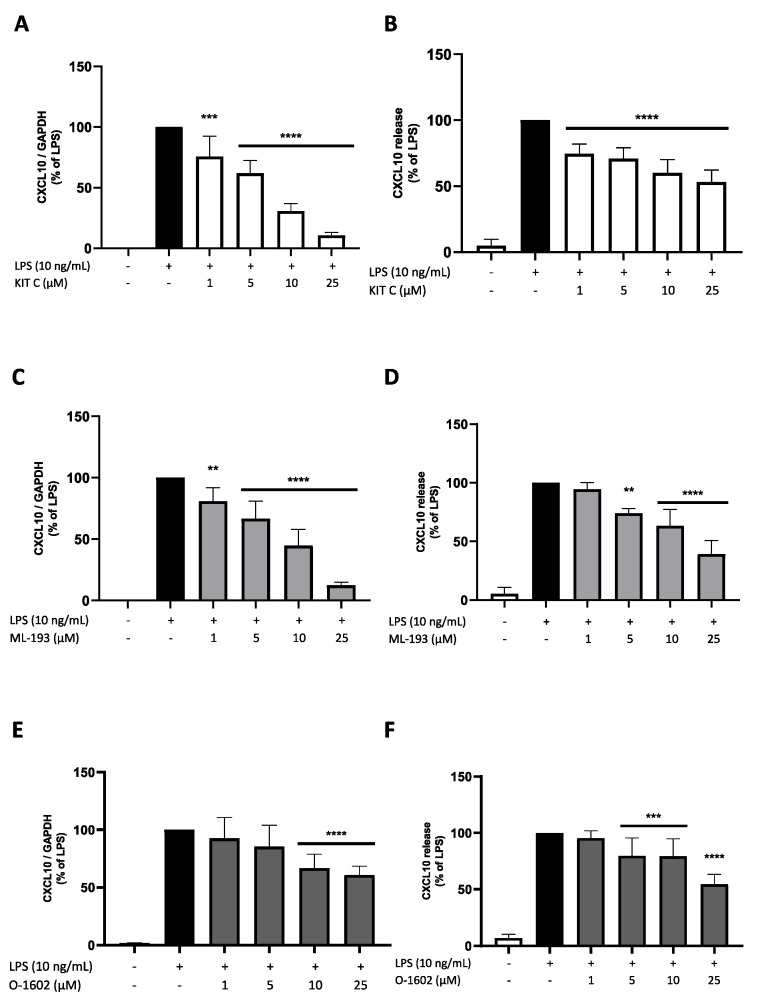
Effects of KIT C (**A**,**B**), ML-193 (**C**,**D**), and O-1602 (**E**,**F**) on the expression (**A**,**C**,**E**) and release (**B**,**D**,**F**) of CXCL10 in LPS-stimulated BV2 cells. Cells were stimulated as described in the Materials and Methods section (4 h for qPCR, 24 h for ELISA). Values are presented as the mean ± SD of at least three independent experiments (N = 4 (**D**), N = 6 (**A**–**C**), N = 7 (**E**), N = 9 (**F**)). Statistical analysis was performed using one-way ANOVA with Dunnett’s post hoc tests with ** *p* < 0.01, *** *p* < 0.001 and **** *p*< 0.0001 compared to LPS-stimulated cells.

**Figure 9 pharmaceuticals-17-00674-f009:**
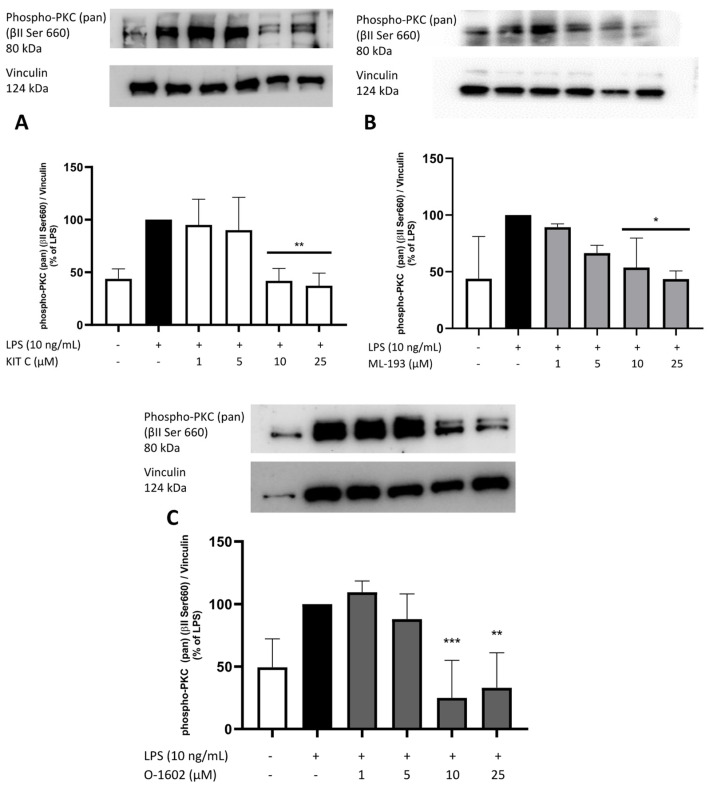
Effects of KIT C (**A**), ML-193 (**B**), and O-1602 (**C**) on phosphorylation of PKC (pan) (βII Ser660) in LPS-stimulated BV2 cells. Cells were stimulated as described in the Materials and Methods section (30 min stimulation). Values are presented as the mean ± SD of at least three independent experiments (N = 3 (**A**,**B**), N = 4 (**C**)), and protein levels were referenced to vinculin. Statistical analysis was performed using one-way ANOVA with Dunnett’s post hoc tests with * *p* < 0.05, ** *p* < 0.01, and *** *p* < 0.001 compared to LPS-stimulated cells. The graphs show an average of the biological replicates and do not directly represent the blot shown above.

**Figure 10 pharmaceuticals-17-00674-f010:**
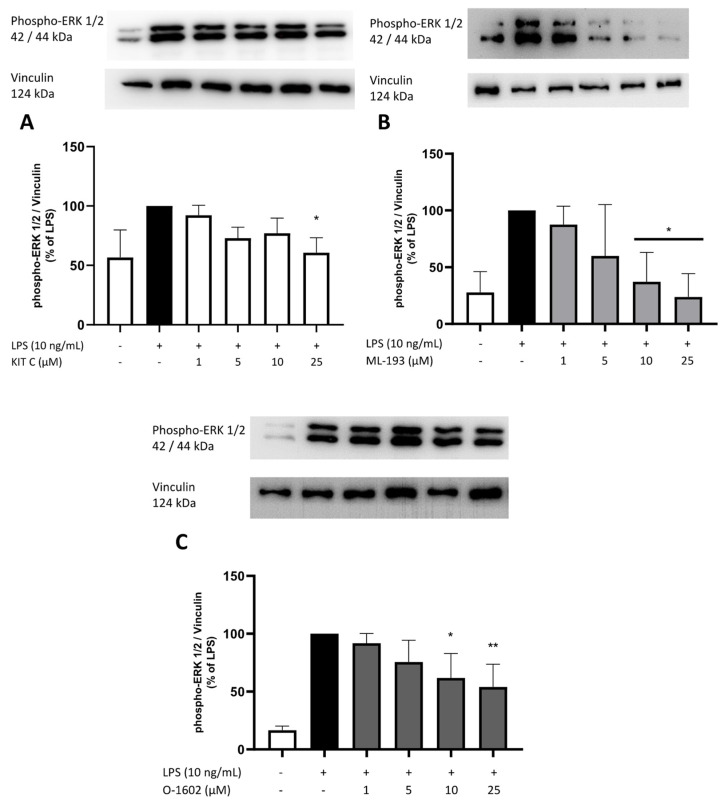
Effects of KIT C (**A**), ML-193 (**B**), and O-1602 (**C**) on the phosphorylation of ERK 1/2 in LPS-stimulated BV2 cells. Cells were stimulated as described in the Materials and Methods section (30 min stimulation). Values are presented as the mean ± SD of at least three independent experiments (N = 3 (**A**–**C**)), and protein levels were referenced to vinculin. Statistical analysis was performed using one-way ANOVA with Dunnett’s post hoc tests with * *p* < 0.05 and ** *p* < 0.01 compared to LPS-stimulated cells. The graphs show an average of the biological replicates and do not directly represent the blot shown above.

**Figure 11 pharmaceuticals-17-00674-f011:**
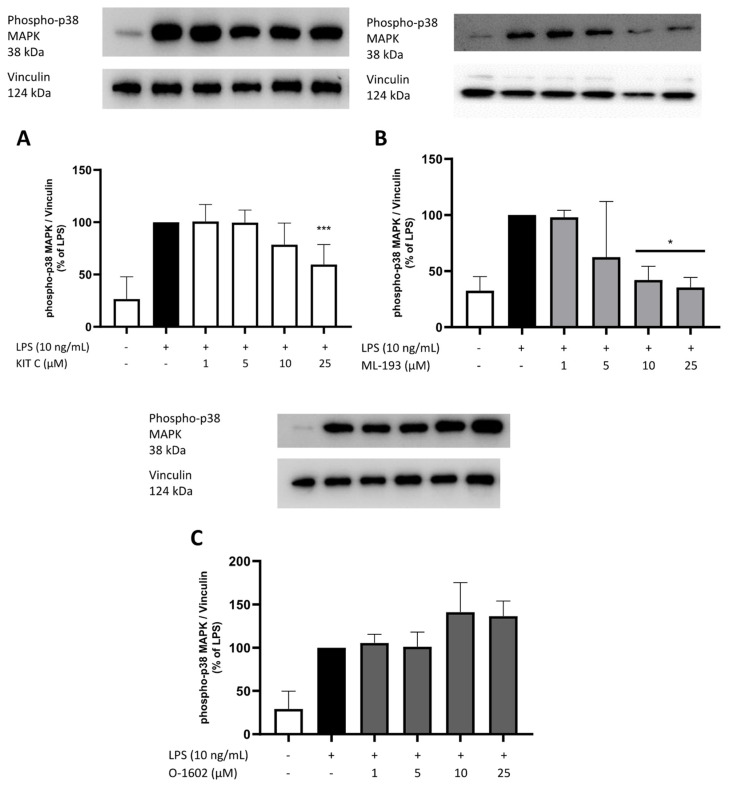
Effects of KIT C (**A**), ML-193 (**B**), and O-1602 (**C**) on the phosphorylation of p38 MAPK in LPS-stimulated BV2 cells. Cells were stimulated as described in the Materials and Methods section (30 min stimulation). Values are presented as the mean ± SD of at least three independent experiments (N = 3 (**B**,**C**), N = 6 (**A**)), and protein levels were referenced to vinculin. Statistical analysis was performed using one-way ANOVA with Dunnett’s post hoc tests with * *p* < 0.05 and *** *p* < 0.001 compared to LPS-stimulated cells. The graphs show an average of the biological replicates and do not directly represent the blot shown above.

**Figure 12 pharmaceuticals-17-00674-f012:**
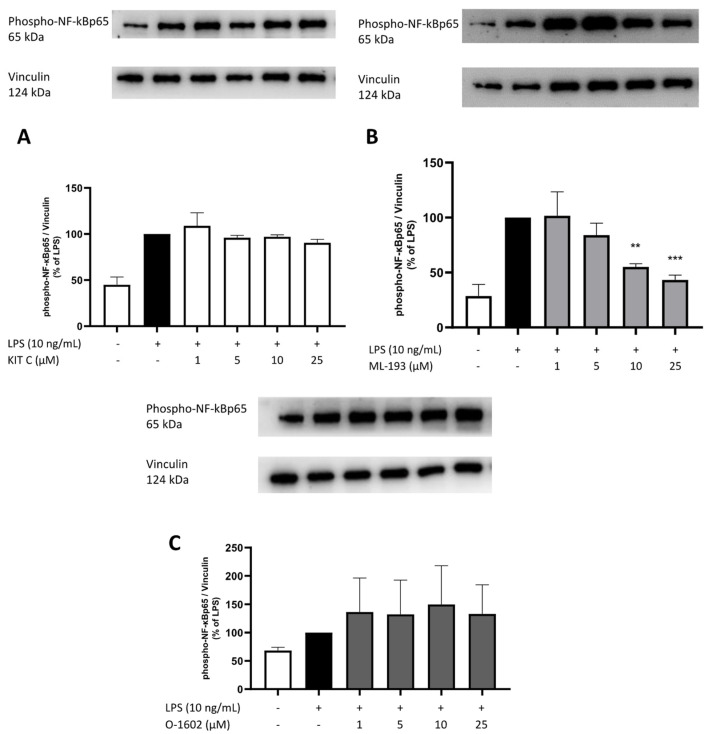
Effects of KIT C (**A**), ML-193 (**B**), and O-1602 (**C**) on NF-κBp65 phosphorylation in LPS-stimulated BV2 cells. Cells were stimulated as described in the Materials and Methods section (30 min stimulation). Values are presented as the mean ± SD of at least three independent experiments (N = 3 (**A**–**C**)), and protein levels were referenced to vinculin. Statistical analysis was performed using one-way ANOVA with Dunnett’s post hoc tests with ** *p* < 0.01 and *** *p* < 0.001 compared to LPS-stimulated cells. The graphs show an average of the biological replicates and do not directly represent the blot shown above.

**Figure 13 pharmaceuticals-17-00674-f013:**
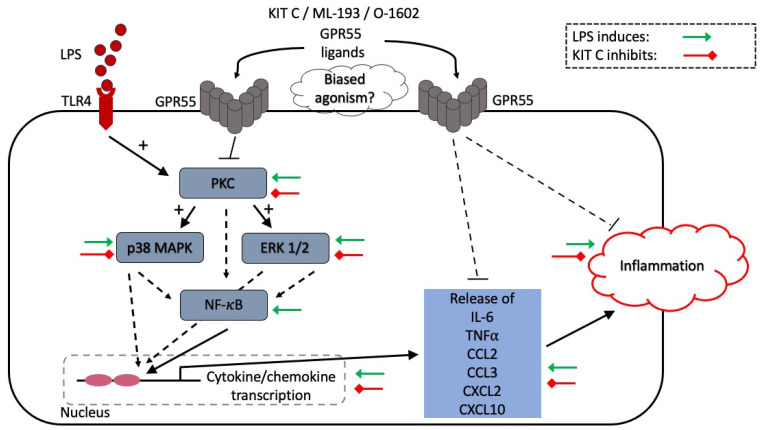
Observed effects of LPS and KIT C in BV2 microglial cells.

## Data Availability

The data presented in this manuscript are available from the corresponding author upon request. Additional information on the synthesis and analysis of the investigated compound KIT C is available via the Chemotion repository (https://www.chemotion-repository.net/ accessed on 18 April 2024) and can be accessed using the following links: https://dx.doi.org/10.14272/LSYXPDGXGOSFCW-UHFFFAOYSA-N.1 (Sample DOI); https://dx.doi.org/10.14272/reaction/SA-FUHFF-UHFFFADPSC-LSYXPDGXGO-UHFFFADPSC-NUHFF-NUHFF-NUHFF-ZZZ (Reaction DOI) [87].

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
