# Peer review of "Anti-Inflammatory Effects of GPR55 Agonists and Antagonists in LPS-Treated BV2 Microglial Cells"

_pharmaceuticals, 2024, doi:10.3390/ph17060674_

Round 1

Reviewer 1 Report

Comments and Suggestions for Authors

The manuscript entitled “Anti-inflammatory effects of GPR55 agonists and antagonists in LPS-treated BV2 microglial cells” by Lu Sun et al., was well written and presented. Authors studies the anti inflammatory role of GPR55 agonists and antagonists in LPS-treated BV2 microglial cells. They used various experiments and analysis to evaluate the anti inflammatory  activities. In particular, the provision of statistical F values is much appreciated.  

However, few queries need to addressed in this manuscript

1. Figure 1. If possible change bar colors to make more understandable

2. Why the authors used vinculin to normalize the western blots?

3. How many replicates used for western blots?

4. Kindly recheck and update this URL: (https://www.chemotion-repopsitory.net/)

5. Figure 10B, western blot bar graphs are average of replicates or representation of that above blot? Band for ML-193 10 and 25 micro molar differs

6. have you checked COX-2, is there any particular reason to left out?

7. If the authors choose to pursue further research in this area, , in vivo experiments could be beneficial.

Author Response

The manuscript entitled “Anti-inflammatory effects of GPR55 agonists and antagonists in LPS-treated BV2 microglial cells” by Lu Sun et al., was well written and presented. Authors studies the anti inflammatory role of GPR55 agonists and antagonists in LPS-treated BV2 microglial cells. They used various experiments and analysis to evaluate the anti inflammatory activities. In particular, the provision of statistical F values is much appreciated. 

However, few queries need to addressed in this manuscript

  1. Figure 1. If possible change bar colors to make more understandable

Response: We believe that the chosen colors give quite a good overview about the different experimental groups. Black bars stand for LPS throughout all figures of the result section. The white bar left to the LPS control are unstimulated cells in all figures. In figure 1, further controls, namely ethanol and DMSO are yellow, to be able to distinguish them from our tested compounds more easily. All compounds are represented by one color in all result figures, KIT C white bars, ML-193 light grey and O-1602 dark grey. However, if you have a concrete idea for color changes, we are looking forward to your suggestions. 

  1. Why the authors used vinculin to normalize the western blots?

Response: Vinculin is a constitutively expressed protein that maintains relatively stable levels across different experimental conditions and cell types. Vinculin levels may remain relatively stable upon activation of NF-kB or MAPK pathways (Carisey & Ballestrem, 2011), as its expression is less likely to be influenced by signaling-induced changes compared to tubulin, GAPDH, or actin.

  1. How many replicates used for western blots?

Response: Most Western Blots were replicated three times. Phospho-PKC (pan) (bII Ser660) was replicated 4 times for O-1602 only. Furthermore, phospho-p38 MAPK was replicated 6 times for KIT C. The exact numbers of replicates are already included in the figures’ legends.

  1. Kindly recheck and update this URL: (https://www.chemotion-repopsitory.net/)

Response: Thank you very much for noticing that typing error. We corrected the link, and it leads to the correct website now.

  1. Figure 10B, western blot bar graphs are average of replicates or representation of that above blot? Band for ML-193 10 and 25 micro molar differs

Response: For all Western Blots, the graph shows the average of replicates after reference to vinculin as percentage compared to LPS. We included a sentence to clarify that in the Western Blot figures legends.

  1. have you checked COX-2, is there any particular reason to left out?

Response: We have checked effects of KIT C, ML-193 and O-1602 on COX-2 (expression and synthesis) in previous papers, focusing on the arachidonic acid/COX-2/mPGES/PGE2 pathway in IL-1b stimulated SK-N-SH cells (Apweiler et al., 2022). Since the current paper shifts the focus on cytokine and chemokine release, we did not assess COX-2 for the recent study.

  1. If the authors choose to pursue further research in this area, , in vivo experiments could be beneficial.

Response: We do agree that future studies should include advanced in vitro/ex vivo experiments, e.g. in organotypic hippocampal slices as tissue model, to assess interactional cell effects as well as in vivo experiments.

Reviewer 2 Report

Comments and Suggestions for Authors

Please add the structures of the substances used in the study to the manuscript. 

Oh, the authors have added the structure of their compound to the very last section. This is not very convenient from the reader's point of view, because it would be good to imagine what is at stake from the very beginning. So my suggestion to add the structures of all three compounds to the introduction remains valid. In addition, it is necessary to take into account the recommendations of IUPAC.

Line 136, the sentence "Concentrations of 1 μM and 5 μM didn't affect cell viability (not shown)." should be re-write for better undestending.

The section "2.2 Effects of KIT C, ML-193, and O-1602 on the expression and release of inflammatory 152 parameters in BV2 microglial cells" does not contain any experimental data and should be revised.

I would also like to suggest that the authors add a schematic representation of the intended effect of KIT C to the Discussion, since visualizations always simplify understanding.

Author Response

Please add the structures of the substances used in the study to the manuscript.

Oh, the authors have added the structure of their compound to the very last section. This is not very convenient from the reader's point of view, because it would be good to imagine what is at stake from the very beginning. So my suggestion to add the structures of all three compounds to the introduction remains valid. In addition, it is necessary to take into account the recommendations of IUPAC.

Response: We moved the structures of all used GPR55 ligands (KIT C, O-1602, ML-193) to the end of the introduction.

Line 136, the sentence "Concentrations of 1 μM and 5 μM didn't affect cell viability (not shown)." should be re-write for better undestending.

Response: We adjusted the mentioned sentence to improve the understanding (line 157-158).

The section "2.2 Effects of KIT C, ML-193, and O-1602 on the expression and release of inflammatory 152 parameters in BV2 microglial cells" does not contain any experimental data and should be revised.

Response: We moved the paragraph to the beginning of the results and included it in the cell viability data paragraph (lines 145-153). We believe that this paragraph gives a good introduction and explanation for the use of the BV2 microglial cell line.

I would also like to suggest that the authors add a schematic representation of the intended effect of KIT C to the Discussion, since visualizations always simplify understanding.

Response: The submitted graphical abstract summarizes the observed effects of KIT C as well as the commercial GPR55 ligands and should give an easy-to-understand visualized overview.

Reviewer 3 Report

Comments and Suggestions for Authors

Sun and colleagues propose a coumarin-based compound as an anti-inflammatory drug. Although interesting results are shown in this manuscript, some major issues should be fixed before its consideration for publication. 
The proposed compound should be tested with LPS concentrations that compromise cell viability as a drug to reduce cell death in inflammatory conditions.

Kit C has been shown to increase cell viability under similar conditions by the same authors in https://doi.org/10.3390/ijms23020959. The release of IL-6 in response to IL-1beta and LPS was studied in this previous publication, with different results. Although these results are compared and discussed, no good explanation is given for these results
Experiment of ERK phosphorylation induction by LPS showed higher basal phosphorylation and no effects of kit C. Since the experiments are not comparable with those of ML-193 and O-1602, it is more correct to repeat them.

Author Response

Sun and colleagues propose a coumarin-based compound as an anti-inflammatory drug. Although interesting results are shown in this manuscript, some major issues should be fixed before its consideration for publication.

The proposed compound should be tested with LPS concentrations that compromise cell viability as a drug to reduce cell death in inflammatory conditions.

Response: Since the recent study focuses on anti-inflammatory effects, especially on cytokines and chemokines level, rather than on anti-apoptotic effects, we did not include higher LPS concentrations. In our opinion, the anti-apoptotic effects should be investigated separately, since different pathways are associated with the anti-apoptotic/pro-proliferative effects of KIT C. Furthermore, the increased cell viability, that was observed in IL-1b stimulated SK-N-SH cells (Apweiler et al., 2022), was not found in LPS-stimulated BV2 microglial cells.

Kit C has been shown to increase cell viability under similar conditions by the same authors in https://doi.org/10.3390/ijms23020959. The release of IL-6 in response to IL-1beta and LPS was studied in this previous publication, with different results. Although these results are compared and discussed, no good explanation is given for these results

Response: There are a couple of possible explanations for the different results. To sum up the findings of our previous study, ML-193 significantly inhibited IL-1b IL-6 release, while KIT C and O-1602 did not affect the IL-1b induced IL-6 release. However, when IL-6 release was examined in LPS-stimulated primary mouse microglial cells, IL-6 release was reduced around 50% by KIT C treatment (Apweiler et al., 2022). These contrary findings were discussed in the mentioned previous study. First, two species are compared (human SK-N-SH cells vs. primary mouse microglial cells). The GPR55 shows species-dependent alterations. While the human GPR55 is formed by 319 amino acids, the murine form contains 327 amino acids. Furthermore, in humans more different splicing forms were observed. Second, two different types of cells (neuronal vs. microglial cells) are compared. Due to different receptor density on cell surfaces, distinct cell/tissue-dependent effects have been observed for the same receptor and ligands before. Such effects have been described for carbachol on muscarinic GPCRs (Kenakin, 2007) for example. Third, biased agonism might be explained by ligand-specific transducer activation of GPCRs. Last, the stimulation with IL-1b and LPS does trigger alternative pathways, that might explain differentiated effects of the compounds in the comparison. We included additional information in the discussion (lines 524-528).

Experiment of ERK phosphorylation induction by LPS showed higher basal phosphorylation and no effects of kit C. Since the experiments are not comparable with those of ML-193 and O-1602, it is more correct to repeat them.

Response: We observed a significant effect of KIT C on the LPS-induced phosphorylation of ERK 1/2 at a concentration of 25 µM. The basal phosphorylation is higher compared to the experiments for ML-193 and O-1602, however, we observed a higher SD of the negative control. We still believe, that the shown experimental results for the phosphorylation of ERK 1/2 for KIT C are reliable and valid without further biological replicates.

Reviewer 4 Report

Comments and Suggestions for Authors

Manuscript by Fiebich et al. reports the anti-inflammatory effects of GPR55 agonists and antagonists in LPS-treated BV2 microglial cells. In this study, authors have identified the anti-inflammatory effects of KIT C (the coumarin-based compound), acting as an antagonist with inverse agonistic activity at GPR55, in comparison to the commercial GPR55 agonist O-1602 and antagonist ML-193.

I would recommend the manuscript for publication after following revisions.

1.     The the structure of coumarin derivate KIT C should be provided at the end of introduction section rather than at the end of manuscript.

2.     The structure of other commercial compounds O-1602 and ML-193 should also be mentioned in the manuscript.

3.     In each figure, authors should mention the time-period of that experiment.

4.     Authors should mention that why did they select KIT C for this study?

5.     In Fugire 10 C, why O-1602 does not change p38 MAPK phosphorylation significantly.

6.     In Figure 11 A, author should provide the reason that why KIT C does not inhibit the LPS-induced phosphorylation of NF-κBp65 however ML-193 do.

Author Response

Manuscript by Fiebich et al. reports the anti-inflammatory effects of GPR55 agonists and antagonists in LPS-treated BV2 microglial cells. In this study, authors have identified the anti-inflammatory effects of KIT C (the coumarin-based compound), acting as an antagonist with inverse agonistic activity at GPR55, in comparison to the commercial GPR55 agonist O-1602 and antagonist ML-193.

I would recommend the manuscript for publication after following revisions.

  1. The the structure of coumarin derivate KIT C should be provided at the end of introduction section rather than at the end of manuscript.

Response: We moved the structure of KIT C to the introduction section.

  1. The structure of other commercial compounds O-1602 and ML-193 should also be mentioned in the manuscript.

Response: We added the structures of the commercial GPR55 ligands O-1602 and ML-193 to the introduction section beside the KIT C structure.

  1. In each figure, authors should mention the time-period of that experiment.

Response: We added LPS stimulation times in all figures’ legends in the results section. For the exact experimental procedure (pre-treatment with the compounds, followed by LPS-stimulation) we refer to the methods and materials section.

  1. Authors should mention that why did they select KIT C for this study?

Response: KIT C was chosen, since it showed more promising anti-inflammatory effects, measured by IL-6 and PGE2 release in LPS-stimulated primary mouse microglia in our previous study (Apweiler et al., 2022). We included a sentence in the introduction to address your question (lines 132-134).

  1. In Fugire 10 C, why O-1602 does not change p38 MAPK phosphorylation significantly.

Response: We can’t provide a reason, why O-1602 does not significantly change the LPS-induced phosphorylation of NF-kB like ML-193 does. However, we observed a non-significant increase of p38 MAPK phosphorylation in concentrations of 10 and 25 µM. Therefore, the agonist O-1602 shows at least a tendency of an opposite effect as compared to ML-193. A possible explanation might be a different activation of distinct signaling pathways via GPR55 dependent on the ligand. This effect is referred to as biased agonism, which occurs due to ligand-specific GPCR conformations and ligand-specific GPCR transducer activation. Additionally, receptor-independent pathways should be taken into account as another possible explanation. In the discussion, we mentioned those possibilities (lines 524-528)

  1. In Figure 11 A, author should provide the reason that why KIT C does not inhibit the LPS-induced phosphorylation of NF-κBp65 however ML-193 do.

Response: We can’t provide a reason, why KIT C does not inhibit the LPS-induced phosphorylation of NF-kB like ML-193 does. However, we hypothesize that biased agonism might be relevant for the observed differences in the effects on NF-kB phosphorylation. Biased agonism means a ligand-distinct biological response, e.g. due to alternated GPR55 transducers activated by the ligand. Different distinct GPCR conformations are responsible for biased agonisms and are described in various cells and for multiple GPCRs already (e.g. (Gay et al., 2004). Another additional explanation are receptor-independent pathways/effects by the different compounds, that may vary.

Round 2

Reviewer 2 Report

Comments and Suggestions for Authors

The authors made some changes to the manuscript, but the authors ignored my comment about the need for an additional scheme illustrating the reasoning in the Discussion. I would like to note that a graphical abstract cannot serve as a substitute, because firstly, it is not available to me for revision, and secondly, it need reflect both the course of experiments and methods, and not the current state of knowledge about the interaction of signaling molecules and effectors in cascades of biochemical reactions.

The authors added the structures of the compounds at the end of the Introduction, however, the chemical structures of the compounds are not depicted in accordance with the IUPAC. This should be fixed.

Author Response

The authors made some changes to the manuscript, but the authors ignored my comment about the need for an additional scheme illustrating the reasoning in the Discussion. I would like to note that a graphical abstract cannot serve as a substitute, because firstly, it is not available to me for revision, and secondly, it need reflect both the course of experiments and methods, and not the current state of knowledge about the interaction of signaling molecules and effectors in cascades of biochemical reactions.

Response: We included an additional figure in the discussion summarizing the observed effects of LPS and KIT C on the investigated parameters in BV2 microglial cells.

The authors added the structures of the compounds at the end of the Introduction, however, the chemical structures of the compounds are not depicted in accordance with the IUPAC. This should be fixed.

Response: We revised the molecular structures of O-1602 and ML-193 according to the IUPAC.

Reviewer 3 Report

Comments and Suggestions for Authors

The authors meet all the requirements and the manuscript is now ready for submission.

Author Response

The authors meet all the requirements and the manuscript is now ready for submission.

Response: Thank you for your positive feedback.

Round 3

Reviewer 2 Report

Comments and Suggestions for Authors

The revised manuscript can be published